# Modified Bamboo Charcoal as a Bifunctional Material for Methylene Blue Removal

**DOI:** 10.3390/ma16041528

**Published:** 2023-02-11

**Authors:** Qian Liu, Wen-Yong Deng, Lie-Yuan Zhang, Chang-Xiang Liu, Wei-Wei Jie, Rui-Xuan Su, Bin Zhou, Li-Min Lu, Shu-Wu Liu, Xi-Gen Huang

**Affiliations:** 1Key Laboratory of Chemical Utilization of Plant Resources of Nanchang, College of Chemistry and Materials, Jiangxi Agricultural University, Nanchang 330045, China; 2Technical Center of Nanchang Customs, Nanchang 330038, China

**Keywords:** bamboo charcoal, methylene blue, adsorption, degradation, kinetics

## Abstract

Biomass-derived raw bamboo charcoal (BC), NaOH-impregnated bamboo charcoal (BC-I), and magnetic bamboo charcoal (BC-IM) were fabricated and used as bio-adsorbents and Fenton-like catalysts for methylene blue removal. Compared to the raw biochar, a simple NaOH impregnation process significantly optimized the crystal structure, pore size distribution, and surface functional groups and increase the specific surface area from 1.4 to 63.0 m^2^/g. Further magnetization of the BC-I sample not only enhanced the surface area to 84.7 m^2^/g, but also improved the recycling convenience due to the superparamagnetism. The maximum adsorption capacity of BC, BC-I, and BC-IM for methylene blue at 328 K was 135.13, 220.26 and 497.51 mg/g, respectively. The pseudo-first-order rate constants *k* at 308 K for BC, BC-I, and BC-IM catalytic degradation in the presence of H_2_O_2_ were 0.198, 0.351, and 1.542 h^−1^, respectively. A synergistic mechanism between adsorption and radical processes was proposed.

## 1. Introduction

Nowadays, organic wastewater discharge has caused very serious world environmental problems with the rapid development of chemical production in the textile, paper, plastic, rubber, and cosmetics industries. Methylene blue (MB) dye is a commonly used organic pollutant, which has high stability and low biodegradability and is thus difficult to degrade [1]. Various strategies, such as membrane filtration [2], advanced oxidation process (AOPs) [3], photocatalysis [4,5,6], and adsorption [7] have been reported for the treatment of organic wastewater. Among them, adsorption and the heterogeneous Fenton-like method, an AOP based on the activation of H_2_O_2_ by a solid catalyst, are gaining importance in dye wastewater decontamination in recent excellent reviews due to their advantages of simple operation, high removal efficiency, low operating cost, and good recycling performance [8,9]. 

Biochar is carbon-enriched solid residues commonly produced by biomass carbonization at high temperatures under hypoxic or anaerobic conditions without complex activation processes [10]. Biochar is well-known in wastewater treatment applications due to its abundant pore structure and surface oxygen-containing functional groups [11]. However, the raw biomass-derived biochar always possesses low specific surface area and poor adsorption capacity. For example, Ji et al. reported that fallen leaf-derived biochar had an adsorption capacity for MB of 78.6 mg/g [12]. Sun et al. prepared biochars by pyrolysis at 673 K from eucalyptus, palm bark, and anaerobic digestion residues, with adsorption capacities for MB of 2.0, 2.95, and 9.77 mg/g, respectively [13]. Therefore, modification of carbon materials such as chemical activation, functional surfaces by surface grafting, or deposition with metal nanoparticles via impregnation is an efficient approach to improve adsorption capacity [14,15,16]. However, in the chemical activation reported in literature, biochar is often soaked with acid, alkali, and salt solution and then calcined at high temperatures [17,18,19,20]. Recently, Xu et al. found that only impregnating bamboo char with HNO_3_ without secondary high temperature calcination can improve its adsorption of mercury by introducing functional groups onto the surface of the biochar [21]. Wu et al. explored the effects of acid/alkali (HCl, HNO_3_ or NaOH) impregnation on the physicochemical properties and adsorption behavior of sludge-based carbon [22] providing a new energy-saving and convenient idea for improving the properties of biochars. On the other hand, biochar is a good support media to disperse metal oxides [23]. Magnetization of Fe_x_O_y_ particles onto biochars has attracted much attention due to its easy recovery performance and good adsorption capacities for organic contaminants [24,25] or heavy metals [26]. However, Liu et al. has pointed out that a large number of uncertainties still need to be explored, such as the synthesis route and potential applications [27].

Reports of biochar for wastewater treatment by AOPs are increasing year by year. Khataee et al. found that Cu_2_O–CuO@BC has better photocatalytic degradation efficiency of RO29 than that of single-component Cu_2_O–CuO and biochar [28]. Zhai et al. used TiO_2_/MgO/ZnO@BC for methylene blue oxidation and found that it can be a bi-functional adsorption and catalytic material with excellent performance [29]. Mian et al. had also reported that MnO_x_–N@BC complexes showed good activity and stability for dye degradation in the presence of PMS [30]. Therefore, loading metal oxides on the surface of biochar can increase new active components and promote catalytic activity. Magnetic biochar–iron oxide composites can not only increase the surface active sites but also promote the convenience of recovery. However, only 8.4% of the published papers about magnetic carbon applications are related to dye removal, and most of them are processed by physical adsorption [31]. Thus, it is necessary to investigate the adsorption and catalytic ability of magnetic biochar in the removal of dye from wastewater. 

Bamboo is a cheap and fast growing woody plant biomass resource and is a sustainable and suitable source of C-based raw materials [32]. In this paper, raw bamboo charcoal, NaOH-impregnated bamboo charcoal, and magnetic modified bamboo charcoal were successively fabricated and used as bio-adsorbents and Fenton-like catalysts for MB removal. Then, the adsorption kinetics, isotherm, and thermodynamics were analyzed and compared in detail. The catalytic activity and reusability as Fenton-like catalysts were studied and a mechanism was proposed. The object of this work is to explore the modified preparation strategy on the pore texture, surface chemistry, and adsorption/catalytic degradation capacities of bamboo charcoal. 

## 2. Materials and Methods

### 2.1. Materials 

The bamboo (Phyllostachys edulis) were taken from Nanchang, Jiangxi province. Sodium hydroxide (NaOH, 96%, AR), hydrogen peroxide (H_2_O_2,_ 30 wt.%, AR), methylene blue (C_16_H_18_CIN_3_S, MB, 98.5%, AR), ferrous chloride tetrachloride (FeCl_2_·4H_2_O, 99%, AR), and ferric chloride hexahydrate (FeCl_3_·6H_2_O, 99%, AR) were purchased from Sinopharm Chemical Reagent Co., Ltd., China. All the chemicals were used without any purification. 

### 2.2. Sample Preparation 

The bamboo were first cut into pieces and subsequently dried for two days at 373 K, and then were crushed into powder. The bamboo charcoal (BC) was produced by calcination of the bamboo powder at 923 K for 2 h under an atmosphere of N_2_. The obtained BC sample was then impregnated with 6 mol/L NaOH solution for 24 h. The impregnated BC sample was then centrifuged, washed, dried, and named BC-I. Next, a certain amount of BC-I (1.15 g), FeCl_2_·4H_2_O (1.837 g) and FeCl_3_·6H_2_O (5 g) were dissolved in 200 mL ultrapure water. The solution was heated to 343 K for 10 min. Then, an NaOH solution was slowly dripped into the mixture and the pH was adjusted to the range of 10~11. The mixture was stirred for 4 h. After filtering, the samples were washed and dried for 24 h at 373 K. The final product was named BC-IM. General preparation mechanism was proposed in Figure 1. 

### 2.3. Sample Characterization 

The crystal phase composition of the three adsorbents were analyzed by XRD (Bruker AXS D8). The pore structures and BET surface were obtained from the adsorption and desorption of N_2_ (Micromeritics TriStar 3000, Micromeritics, USA). The characteristics of the pore size distribution were determined by the Barrett–Joyner–Halenda (BJH) and Horvath–Kawazoe (HK) methods. The morphologies and images of the three adsorbents were investigated by SEM (Philips XL 30, Philips, Czech Republic) and TEM (JEOL 2011, JEOL, Japan), respectively. The element composition analysis and element distribution map of the samples were obtained by EDS coupled to the SEM. FT-IR (Perkinelmer C107996, PerkinElmer, USA) was applied to explore the types of surface functional groups. The magnetic properties were monitored by VSM at room temperature. Finally, XPS (PHI 5000C, PHI, USA) was used to investigate the valence state of surface iron. 

### 2.4. Batch Adsorption Experiments

A total of 100 mg of biochar was poured into 100 mL of a 10 mg/L MB solution at 308 K in a 250 mL glass flask. The adsorption process was investigated by a 722-UV spectrophotometer at 665 nm. Each parameter of the adsorption experiment was repeated three times. The removal efficiency of MB is expressed as (*A*_0_ ‒ *A*)/*A*_0_ × 100%, where *A*_0_ and *A* are the initial absorbance and the absorbance at a given reaction time, respectively. The recycling experiments of the best BC-IM were carried out under similar reaction conditions. The sample was collected by centrifugation, washed, and dried overnight in a vacuum oven at 373 K after each recycle during the recycling experiment. 

Furthermore, a series of adsorption tests were carried out with the initial MB concentration varying from 20 to 50 mg/L and the dosage of adsorbent was 100 mg at 308 K. The influence of adsorption temperature on adsorption properties was also studied at 308, 318, and 328 K. The adsorption capacity *Q_e_* (mg/g) at adsorption equilibrium and *Q_t_* (mg/g) at the selected reaction time were obtained from the following equations:(1)$$Q_{e}=\frac{\left(c_{o}-c_{e}\right)V}{m}$$
(2)$$Q_{t}=\frac{\left(c_{o}-c_{t}\right)V}{m}$$
where *c_o_* (mg/L), *c_e_* (mg/L), and *c_t_* (mg/L) represent the MB concentration at the initial state, adsorption equilibrium, and a selected time *t*, respectively. The *m* (g) and *V* (L) are the mass of adsorbent and the volume of solution used, respectively. 

### 2.5. Catalytic Degradation Experiments

MB degradation by the catalyst/H_2_O_2_ system was used as the model experiment to study the catalytic activity of the prepared catalysts. In a 250 mL conical flask, 0.1 g of catalyst, 20 mL H_2_O_2_, and 100 mL (10 mg/L) MB solution were mixed and placed at 308 K. The absorbance of the solution was measured using a 722-UV spectrophotometer at 665 nm. The catalytic activity of the catalyst was also determined by the removal efficiency of MB, which was calculated as (*A*_0_ ‒ *A*)/*A*_0_ × 100%. The recycling experiments were carried out under similar reaction conditions. After reacting for 2 h, the sample was collected by centrifugation, washed, and dried overnight in a vacuum oven at 373 K after each recycle during the recycling experiment. 

## 3. Results

### 3.1. Characterization Results of the Samples

As demonstrated in Figure 1, the BC, BC-I, and BC-IM samples all exhibited a broad band around 22°, corresponding to the (002) diffraction of the carbon structure, which demonstrates the conversion of lignin–cellulose to a more carbonaceous structure during pyrolysis [33]. For the raw BC, there were many sharp narrow peaks, which may be related to the formation of inorganic ash (various oxides or carbonates) during the pyrolysis process of the bamboo [33,34]. It is worth noting that after NaOH-impregnation, a slight diffraction shift of the (002) plane ranging from 21.7 to 24.5° was observed for the BC-I sample, which has been reported to be related to the formation of more defects [35] and more ordered graphite structures with smaller layer spacing [36]. In addition, a peak around 2*θ* value of 43° was also found for the BC-I sample, which belongs to the (100) diffraction plane of carbons [37]. The appearance of the (100) plane had been reported to be evidence of the formation of graphitic structures [35], which is consistent with the red shift phenomenon of the (002) plane. Therefore, it indicates that the simple NaOH-modification treatment is an efficient way to optimize the crystal structure of biochar materials. After further magnetization, a diffraction peak centered at 35.6° assignable to the iron oxide Fe_2_O_3_ was observed, which indicates that iron oxide was successfully introduced to the surface of bamboo charcoal through a facile one-pot in situ precipitation. Rietveld analysis showed that the Fe_2_O_3_ (PDF-39-1346) is a cubic system with a cell parameter of 0.8832 nm.

Figure 2 presents the SEM images and EDS patterns of the three adsorbents, as well as the elemental mappings of the BC-IM sample. As shown in the SEM figures, BC-I had a rough surface and a large number of irregular channels and folds compared to the raw BC, providing a wide field for the adsorption of organic pollutants [25]. The SEM image of BC-IM showed not only more pore structures but also a large number of particle aggregates distributed around the pores, indicating the formation of magnetic particles on the surface of BC-IM. To identify the differences in the elemental composition of the adsorbents, the EDS patterns were also displayed. The results showed that the raw BC sample contained C, O, K, and Cl elements, with estimated atomic contents of 88.29%, 9.44%, 1.99%, and 0.28%, respectively. The element composition of the BC-I sample was C (89.43%), O (9.16%), and Na (1.41%). The composition of BC-IM was C (33.02%), O (45.61%), Na (3.41%), Cl (58.6%), and Fe (17.58%), which proves that iron was successfully loaded onto the surface of the bamboo charcoal. The elemental mappings of the BC-IM sample further displayed the distribution of iron on the surface of the BC-IM.

The nitrogen adsorption–desorption isotherms and corresponding pore size distribution curves of BC-I and BC-IM are summarized in Figure 3. The pore size, pore volume, and BET surface area are displayed in Table 1. For the raw BC sample, a low BET surface area value of 1.4 m^2^/g and small pore volume of 0.001 cm^3^/g were obtained. The above XRD results demonstrated that the raw BC sample contained some inorganic ash, which may cover the pore structure and surface active sites of BC. The isotherms of both BC-I and BC-IM samples exhibited a rapid nitrogen uptake at the low relative pressure region of less than 0.06, followed by a continuous increase in the rest relative pressure with a large hysteresis loop. In general, the relative pressure in the adsorption–desorption isotherm at 0~0.1, 0.1~0.8, and 0.8~1 is regarded as associated with micropores, mesopores, and macropores, respectively [25]. It shows that high proportion of micropores accompanied with partial mesopores or even macropores appeared in the BC-I and BC-IM samples, which is believed to be beneficial for the transport and diffusion of dyes during adsorption [35].

The detailed pore characteristics can be further observed by the pore size distribution curves. The BC-I sample contained pores 0.7–17 nm in size with concentrated pore sizes of 0.8, 1, and 6.2 nm. The BET surface area and pore volume of BC-I were 63.0 m^2^/g and 0.082 cm^3^/g, respectively. Li et al. had also synthesized a series of bamboo hydrochars with BET surface areas ranging from 2.6 to 43.1 m^2^/g [38]. Therefore, it is obvious that the impregnation method using NaOH can effectively remove inorganic ash in the raw BC sample and improve the surface area and pore structure. After the magnetizing treatment, the BC-IM showed a large number of micropores centered at 0.9 nm, partial mesopores centered at 3.8 nm, and small number of macropores. The dimensions of the MB molecule in water is reported as 0.400 × 0.793 × 1.634 nm; Santoso et al. reported that micropores and mesopores less than 6 nm are crucial to increase the amount of adsorbed MB molecules [39]. Furthermore, the BC-IM sample had a higher specific area (84.7 m^2^/g) and larger pore volume (0.112 cm^3^/g). Therefore, a more suitable pore size, higher surface area, and larger pore volume will make the BC or BC-IM samples have more excellent adsorption capacity for MB. 

Figure 4 shows the FT-IR diagrams of BC, BC-I, and BC-IM. Compared to the raw BC, the stretching vibration peak of O-H at 3497 cm^−1^ and C=C bond of the vibration of the sp^2^ carbon skeletal network at 1640 cm^−1^ were significantly enhanced in the NaOH-impregnated BC-I sample [40]. After magnetization, more surface functional groups appeared. The obvious bands between 1408 and 1662 cm^−1^ can be ascribed to C=O stretching modes in aromatic ring structures or ring vibrations in a large aromatic skeleton [41,42]. In addition, the band around 760 cm^−1^ is related to the C-C and C-H stretching vibration in aromatic rings [43]. The band at 1020 cm^−1^ is attributed to the lactone structure of C-O-C [25]. The unique band observed at about 517 cm^−1^ in the BC-IM sample is assigned to the stretching vibration of Fe-O, which proves the appearance of iron oxide [44]. This indicates that the oxygen-containing functional groups on the surfaces of BC-I and BC-IM can be greatly increased by modification. 

TEM, HRTEM, and SAED images and VSM magnetization curves of the BC-IM sample are shown in Figure 5. Apparently, there are many agglomerated particles on the surface of the rod-like bamboo charcoal structure with diameter of about 65 nm, owing to the load of iron oxide (Figure 5a). The d-spacing of 0.649 nm was observed in the HR-TEM patterns, which corresponds to the (200) lattice planes of the Fe_2_O_3_ phase. It proves that Fe_2_O_3_ was successfully loaded on the surface of BC-IM, which is in accordance with the XRD results. The corresponding SAED pattern exhibited a polycrystalline structure pattern. The hysteresis loop at room temperature of BC-IM is shown in Figure 6d. The absence of residual matter and coercivity in the magnetic ring proves that the synthesized BC-IM is superparamagnetic. In addition, the saturation magnetization of BC-IM was 16.4 emu/g. It shows that BC-IM can be easily separated from the solution [45].

The XPS technique was also employed to explore the surface species of BC-IM (Figure 6). The C1s spectrum can be fitted into four peaks. The binding energy peak at 288.8 eV of C1s contains both CO_3_^2−^ and –COOH(R) oxidation states. The peaks at 286.5, 285.0, and 284.5 eV belong to the functional groups C=O, C-O and C-C, respectively [21,22]. The O1s spectrum can be fitted into five peaks located at the binding energy of 530.05, 530.6, 531.6, 533.2, and 537.0 eV, which are ascribed to Fe-O-Fe, C-O, C=O/Fe-OH, COOH(R), and O/H_2_O functional groups [46,47]. All the above analysis indicates that there are abundant functional groups in BC-IM, which is consistent with the FT-IR results. Next, the Fe2p peaks could be deconvoluted into some characteristic peaks, among which the four peaks at 726.0, 724.4, 712.9, and 711.2 eV belong to Fe^3+^ and the two peaks at 723.3 and 710.1 eV belong to Fe^2+^ [47,48]. The results showed that BC-IM has multivalent iron elements.

### 3.2. Adsorption Properties of Samples

The Performances of the synthesized adsorbents were measured for the adsorption of methylene blue. As can be seen from Figure 7a, the adsorption capacity of the BC-IM composites was superior to that of BC-I and BC. For BC-IM, the MB removal efficiency reached 83% after 11 h, which was higher than that of BC-I (69%) and BC (60%). MB adsorption by the three adsorbents increased rapidly in the first 10 h, then decreased gradually until the equilibrium state was reached at around 58 h. The removal efficiency at equilibrium of BC, BC-I, and BC-IM were 98%, 93%, and 88%, respectively. Combined with the above characterization results, impregnation and magnetization can enlarge the specific surface area of bamboo charcoal, improve the pore structure, and form more surface oxygen-containing functional groups, which may be favorable factors for the improvement of adsorption capacity. Figure 7a shows the images of the solutions before and after the reaction, showing that BC-IM has a good ability to treat methylene blue in water and can be easily recovered under external magnetic action. As can be seen from the figure, the solution was relatively clear after reaching the adsorption equilibrium, and the BC-IM sample could be recovered by an external magnetic field.

The recycling of biochar used for MB removal was investigated using BC-IM and the results shown in Figure 7b. The adsorption amounts were gradually decreased during the cycling process, which may be attributed to the blockage of pores by adsorbed MB molecules. The removal efficiency of 80% after five cycles showed that the BC-IM sample can act as a potential adsorbent for MB removal. 

The point of zero charge (pHpzc) of samples can be determined using a previously reported method [7]. As can be seen in Figure 8, the pHpzc of the three adsorbents were all around 9.4. This indicates that at pH above 9.4, the surface of the adsorbent is considered to have a positive charge, which can adsorb cationic species (like MB). Therefore, the effect of pH on MB removal by BC-IM was examined by varying the pH values from 5 to 12. The initial MB concentration was 10 mg/L at room temperature and reaction lasted for 24 h. At pH > pHpzc, there was an electrostatic attraction between the negative loads from the adsorbent surface and the positive loads from the MB molecules. Therefore, higher removal of MB was achieved in the pH range of 10–12 (Figure 8d). Meanwhile, the electrostatic repulsion, chemical reaction, and Fe leaching in acidic conditions led to the very poor adsorption efficiency of BC-IM. 

### 3.3. Adsorption Kinetics, Adsorption Isotherm, and Adsorption Thermodynamics

To explain the MB adsorption mechanism, pseudo-first-order, pseudo-second-order, and intra-particle diffusion models were first applied to explain the MB adsorption kinetics using the raw BC, BC-I, and BC-IM samples. The equations of these kinetic models are as follows [22,43,49]:(3)$$\mathrm{lg}\left(Q_{e}-Q_{t}\right)=\mathrm{lg}Q_{e}-k_{1}t$$
(4)$$\frac{t}{Q_{t}}=\frac{1}{k_{2}Q_{e}^{2}}+\frac{t}{Q_{e}}$$
(5)$$Q_{t}=k_{ip}\;t^{1/2}+C_{i}$$
where *k*_1_ (h^−1^) and *k*_2_ (g/(mg·h)) obtained from the linear fitting slope of the equations are the adsorption rate constant of the pseudo-first-order or pseudo-second-order models, respectively. *k_ip_* (mg/(g·h^0.5^)) is the rate constant of the intra-particle diffusion model and *C_i_* (mg/g) represents the intercept reflecting the boundary layer effect. 

The effect of contact time on MB adsorption capacity is shown in Figure 9. It can be seen that adsorption capacity increased with the contact time. The corresponding fitting results and kinetic parameters are shown in Figure 10 and Table 2. A higher correlation coefficient value (*R^2^*) was observed in the pseudo-second-order model for the three adsorbents, indicating that the adsorption system conforms to the pseudo-second-order kinetic model. The basic assumption of the pseudo-second-order model is that chemisorption is the rate-controlled step and chemisorption originates from valence forces between adsorbate and adsorbent by electron exchange or sharing [50]. Therefore, chemisorption is the rate-determining step for MB adsorption, limiting the mass transfer induced participation in the solution [51].

Then, the Langmuir, Freundlich, and Temkin adsorption isotherm models were applied to evaluate the mechanism during the adsorption process using BC, BC-I, and BC-IM at different temperature. In addition, these model equations are as follows [22,52]:(6)$$\frac{c_{e}}{Q_{e}}=\frac{c_{e}}{Q_{m}}+\frac{1}{Q_{m}k_{L}}$$
(7)$$\ln Q_{e}=\ln k_{F}+\frac{1}{n}\;\ln c_{e}$$
(8)$$Q_{e}=B_{T}\ln A_{T}+B_{T}\ln c_{e}$$
where *Q_m_* (mg/g) represents the amount of MB at the complete cover state, which displays the maximum adsorption capacity; and *k_L_* (L/mg) is the Langmuir constant related to the adsorption energy; *k_F_* is the Freundlich constant related to adsorption capability, and 1/*n* represents the adsorption intensity; and *A_T_* (L/mg) represents the maximum binding energy and *B_T_* (mg/g) is the heat change during the adsorption, which are the Temkin model constants.

Adsorption isotherms of MB on the three samples at different temperature are shown in Figure 11. Apparently, MB adsorption capacity of all samples increased as the MB concentration increased, resulting from the increased mass transfer driving force during adsorption. The adsorption isotherms were nearly linear, which suggested a complex adsorption mechanism involving partitioning for the adsorption of MB by the bamboo charcoals [38]. 

The corresponding linear fitting results from the Langmuir, Freundlich, and Temkin models are listed in Figure 12 and Table 3. Compared with the Temkin isotherms, the Langmuir and Freundlich isotherms fit the equilibrium data well. Therefore, the Langmuir and Freundlich models can explain the adsorption mechanism of the three adsorbents. For the raw BC and BC-IM, the Langmuir isotherm model was more suitable for describing the MB adsorption process than the Freundlich model on the whole. It shows that the surface of the BC and BC-IM adsorbents is homogeneous and the adsorption takes place in the monolayer. Meanwhile, for the BC-I sample, the value of *R^2^* of the Langmuir isotherm model became lower than that of the Freundlich model with the increase in adsorption temperature, indicating that the Freundlich model has higher accuracy. Moradi et al. had also reported the transition of the adsorption isotherm to the Freundlich model at higher temperatures [53]. It shows that the heterogeneous surface may play an important role in multi-layer adsorption for MB removal [22]. Moreover, the 1/n values in the Freundlich isotherm varied from 0.1 to 1, which shows that the sorption process is favorable [54]. The maximum adsorption capacity (*Qm*) of BC, BC-I, and BC-IM obtained by the Langmuir isotherm model at 328 K was 135.13, 220.26, and 497.51 m^2^/g, respectively. Compared to the adsorption capacity for MB from numerous adsorbents reported in the literature (36.25~384.61 mg/g) [43], bamboo charcoal can be used as efficient adsorbents of MB from wastewater.

Finally, the thermodynamic parameters of MB adsorption by the three bamboo charcoals including the Gibbs free energy change (Δ*G*^0^), enthalpy change (Δ*H*^0^), and entropy change (Δ*S*^0^) were calculated from the following equations:(9)$$\Delta G^{0}=-RT\ln K_{c}$$
(10)$$K_{c}=\frac{Q_{e}}{c_{e}}$$
(11)$$\ln K_{c}=\frac{\Delta S^{0}}{R}-\frac{\Delta H^{0}}{RT}$$
where *K_c_* (L/g) represents the equilibrium constant for the MB adsorption process. The Δ*H*^0^ andΔ*S*^0^ were evaluated by plotting ln *K_c_* against 1/*T* (Figure 13).

The values of Δ*G*^0^, Δ*H*^0^, and Δ*S*^0^ are summarized in Table 4. All the Δ*G*^0^ values were negative for the three adsorbents at different temperatures, which indicates that MB adsorption is spontaneous and does not require external energy input. Generally, the physical adsorption energy is from 0 to −20 kJ/mol, while that of chemical adsorption is from −80 to −400 kJ/mol [55]. The Δ*G*^0^ values of the three adsorbents shown in Table 4 at different temperatures are in the range of −3.75~−0.97 kJ/mol, indicating that the adsorption can be regarded as physical adsorption. At the same time, the values of Δ*H*^0^ were less than 25 kJ/mol, further proving that the MB adsorption process belongs to physical adsorption. For BC and BC-IM samples, the negative values of Δ*H*^0^ and Δ*S*^0^ showed that MB adsorption is a randomness decrease and exothermic process. Meanwhile, for the BC-I sample, the positive Δ*H*^0^ and Δ*S*^0^ values denotes that the adsorption of MB by BC-I is endothermic and the randomness is raised at the solid–liquid interface. 

The adsorption kinetics suggests that the obtained adsorption data fit best with the pseudo-second order model, which indicates a chemical adsorption. However, it is apparent from the thermodynamics and adsorption isotherm results that physical reaction also occurred in the MB adsorption process. In conclusion, MB adsorption by bamboo charcoals is a complex physical and chemical reaction process. 

### 3.4. Catalytic Properties and Reusability of Samples

The catalysis experiments of the three samples were conducted at 308 K and are summarized in Figure 14a. The removal efficiency of MB in the presence of H_2_O_2_ by BC, BC-I, and BC-IM was 58.85%, 70.95%, and 98.43%, respectively. For comparison, the pure adsorption experiment was also performed. The removal efficiency of MB in the absence of H_2_O_2_ by BC, BC-I, and BC-IM was 38.65%, 42.57%, and 71.46%, respectively. Meanwhile, the degradation efficiency was less than 14% only in the presence of hydrogen peroxide and no catalyst. The significant improvement of removal efficiency after adding hydrogen peroxide indicates that Fenton-like catalysis also plays an important role in methylene blue removal, besides adsorption. Pseudo first-order kinetics was used to obtain the kinetic rates of MB degradation (Figure 14b). The fitted rate constants at 308 K of BC, BC-I, and BC-IM were 0.198 h^−1^, 0.351 h^−1^, and 1.542 h^−1^, respectively. Apparently, the degradation capabilities of the modified bamboo charcoals were higher than that of the raw BC. In particular, the BC-IM sample showed the best rate constants, which was also better than other Fenton-like catalytic systems used in methylene blue degradation reactions (Table 5). 

The reusability of the best sample (BC-IM) is also shown in Figure 14a. The removal efficiency after four cycles was 94.6%, 89.5%, 84.7%, and 79.0%, respectively. The loss of activity may be caused by the slight leaching of iron ions from the catalyst and the surface active site being blocked by byproducts. The obtained results showed that the Fenton process can stably remove most of the MB in dye-polluted waters, suggesting that the BC-IM sample can act as a potential Fenton catalyst for the treatment of actual wastewater.

### 3.5. Mechanism Study

The MB degradation process is proposed to be a synergistic mechanism involving adsorption and radical processes. For the raw BC sample, the MB and H_2_O_2_ molecules are first adsorbed on the surface of BC, the persistent free radicals (PFRs) existed on the biomass charcoal will facilitate the decomposition of H_2_O_2_ into hydroxyl radical [59], which can further attack and degrade MB molecules. Finally, the byproducts produced by degradation are desorbed. As described above, a simple NaOH impregnation process can significantly optimize the crystal structure, pore size distribution, and surface functional groups and increase the specific surface area from 1.4 to 63.0 m^2^/g. Therefore, the enhancement of degradation efficiency is related to these factors, which may produce more active sites. Further magnetization of the BC-I sample by one-pot in situ precipitation can not only continuously improve physicochemical properties and facilitate the recycling convenience, but also introduce Fe oxide as a new activator (Equations (1) and (2)).
Fe^2+^ + H_2_O_2_ → Fe^3+^ + ·OH + ^−^OH(12)
OH+ MB → intermediates → CO_2_ + H_2_O(13)

Therefore, the monolayer adsorbed MB on BC and BC-IM or multilayer adsorbed MB on BC-I would be degraded by the hydroxyl radical originating from PFRs and Fe oxide-activated H_2_O_2_. Then, the active sites on the surface of BC-IM can be reused to adsorb MB molecules again, which could improve the MB removal capacity. Magnetic bamboo charcoal can offer a potential opportunity in the field of organic pollutants treatment.

## 4. Conclusions

In this study, a simple and feasible modified strategy of bamboo charcoal without secondary high temperature pyrolysis was studied. In particular, the magnetic modification not increases the maximum sorption capacity and degradation rate constants of BC-IM 3.7 times or 7.8 times that of the raw BC, but also makes it more convenient to quickly recover under an external magnetic field. The pseudo-first-order rate constants *k* at room temperature for BC, BC-I, and BC-IM for catalytic degradation in the presence of H_2_O_2_ were 0.198, 0.351, and 1.542 h^−1^, respectively. A synergistic mechanism involving adsorption and radical processes was proposed, which is expected to act as an adsorption/catalytic bifunctional catalyst in organic pollutants treatment. The findings provide insight into the MB adsorption/catalytic degradation mechanism by bamboo charcoal, and gives guidance to the design of modification strategies for biochars.

## Data Availability

Not applicable.

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
