# Peer review of "Modified Bamboo Charcoal as a Bifunctional Material for Methylene Blue Removal"

_materials, 2023, doi:10.3390/ma16041528_

Round 1

Reviewer 1 Report

This manuscript cannot be published in this journal at the present form and serious major revisions are necessary.

1-     The size of the writing in page 3 is different to the other pages.

2-     section 2.4 and 2.5: absorbance is I0 or A0

3-     BET analyzes showed low specific surface ?? how to interpret that? after modification of bambo by iron oxide the specific surface is increased which is unexpected, normally the particles of Fe2O4 must occupy the pores of the surface of Bambo and the specific surface must be decreased.

4-     the equilibrium time is very large, going up to 58 h!!! which is not practical as an adsorbent on a large scale, especially the initial concentration is very low 10mg/l.

5-     the interpretation of the adsorption models is meaningless, the adsorbed quantity calculated for each model must be added and compared with the experimental adsorbed quantity. the model choice based on R2 is incorrect.

6-     what are the initial concentrations used for the tests of the adsorption isotherms?

7-     Add the figures of the adsorption models Qe and Ce before presenting the linearization of each adsorption model.

8-     add the experimental adsorbed quantity in table 3.

9-     add figure of quantity adsorbed as a function of temperature

10-  how to calculate constant K in figure 11

11-  The authors invited to determine the pH of the zero point of charge of their materials

12-     How many adsorption-desorption cycle can this material use to remove MB dye. The regeneration study is essential in adsorption studies.

13-   What is the effect of pH on MB dye removal?

14-  Manuscripts should refer to and cite as much as possible from the last five years. Some high-quality literatures about sustainability of water in recent years can be referenced and cited, such as : 

https://doi.org/10.1007/s13369-022-06899-y

https://doi.org/10.1016/j.molliq.2021.116560

Reviewer 2 Report

Materials-2150508

I reviewed your manuscript “Facile preparation of modified bamboo charcoal and highly efficient adsorption and catalytic degradation ability for methylene blue” very carefully. The work carried out in the manuscript is interesting and seems logical. The authors have added good technical value to the field and the readers will benefit. However, there are several errors in this work and before publication, it needs to restructure the research manuscript properly as the current presentation is not acceptable. Therefore, I would like to recommend this manuscript for "Major Revision".

1.    The title is too long. It should be short, meaningful and attractive. Please modify the title.

2.    The authors not write the paper diligently. Even mistakes in the abstract section (line 19-20, line 36).

3.    The novelty of this work was not specified; authors should discuss the novelty of their work in the introduction section. The author should made comparison between their photocatalysts and already reported photocatalysts like hexaferrites materials. The author should also read the research articles related to the wastewater purification, chemical physics letters 805, (2022), 139939, 431–440, New J. Chem., 2022,46, 19848 and doi.org/10.1080/03067319.2022.2032014

4.    The percentage purity of all the chemicals utilized must be reported.

5.    The authors should add the flow sheet diagram to show the mechanism of formation of bamboo samples. See Journal of Materials Science: Materials in Electronics 32(7104):1-14 for this purpose

6.    Font size and style issue in line 114, 115, 120-127

7.    I have big question mark on the XRD analysis of the samples. The authors not reported what type of structure exist in the samples. After the addition of iron oxide the sample should show crystallinity while here it showing the amorphous?  I strongly recommend to provide Rietveld analysis of all the samples because photocatalytic properties of the synthesized samples are highly sensitive to the purity of the samples.

8.      The SEM analysis should include immediately after XRD analysis. No connection is present between the XRD and SEM results. I recommend to add all these information in the manuscript. Is there any other method r technique to confirm the elemental analysis more precisely? XRD and SEM.TEM section lacks enough reference.

9.    The authors perform adsorption and degradation experiments at same time. They should explain which experiment was conducted first? Or different samples were used for both experiments? Comparison table between adsorption and photodegradation should added. Moreover, comparison should be made between already reported and your work.

10. Another important parameter (band gap) is also ignored in the manuscript. Please add band gap determination also.

11. I suggest the author to add the digital Photographs of all adsorption and degradation experiments (before degradation and after degradation photograph).

12. Lot of literature available where the degradation and adsorption of dyes has been reported in similar duration, therefore the novelty ion this work is missing. The author should explain the novelty in proper way.

13. Please read the manuscript diligently and remove all the typographical, font style and font size mistakes

14. Revise the conclusion section and add meaningful and numerical values to make it more attractive and easy understanding for new researchers.

Round 2

Reviewer 1 Report

The revised version can be accepted for publication 

Reviewer 2 Report

Accept